# Development of wheat—*Hordeum chilense* Chromosome 2H<sup>ch</sup> Introgression Lines Potentially Useful for Improving Grain Quality Traits

Carmen Palomino and Adoracion Cabrera *

Departamento de Genética, Escuela Técnica Superior de Ingeniería Agronómica y de Montes, Edificio Gregor Mendel, Campus de Rabanales, Universidad de Córdoba, CeiA3, ES-14071 Córdoba, Spain
* Correspondence: acabrera@uco.es

**Abstract:** The chromosome 2H$^{ch}$ of *Hordeum chilense*. has the potential to improve seed carotenoid content in wheat as it carries a set of endosperm carotenoid-related genes. We have obtained structural changes in chromosome 2H$^{ch}$ in a common wheat (*Triticum aestivum* L. "Chinese Spring") background by crossing a wheat double disomic substitution 2H$^{ch}$(2D) and 7H$^{ch}$(7D) line with a disomic addition line carrying chromosome 2C$^c$ from *Aegilops cylindrica* Host.. Seven introgressions of chromosome 2H$^{ch}$ into wheat were characterized by fluorescence in situ hybridization (FISH) and DNA markers. Chromosome-specific simple sequence repeats (SSRs) were used for identifying wheat chromosomes. In addition, we tested 82 conserved orthologous set (COS) markers for homoeologous group 2, of which 65 amplified targets in *H. chilense* and 26 showed polymorphism between *H. chilense* and wheat. A total of 24 markers were assigned to chromosome 2H$^{ch}$ with eight allocated to 2H$^{ch}$S and 16 to 2H$^{ch}$L. Among the seven introgressions there was a disomic substitution line 2H$^{ch}$(2D), a ditelosomic addition line for the 2H$^{ch}$L arm, an isochromosome for the 2H$^{ch}$L arm, a homozygous centromeric 2H$^{ch}$S·2DL translocation, a double monosomic 2H$^{ch}$S·2DL plus 7H$^{ch}$S·D translocation, a homozygous centromeric 7H$^{ch}$S·2H$^{ch}$L translocation and, finally, a 2H$^{ch}$L·7H$^{ch}$L translocation. Wheat—*H. chilense* macrosyntenic comparisons using COS markers revealed that *H. chilense* chromosome 2H$^{ch}$ exhibits synteny to wheat homoeologous group 2 chromosomes, and the COS markers assigned to this chromosome will facilitate alien gene introgression into wheat. The genetic stocks developed here include new wheat—*H. chilense* recombinations which are useful for studying the effect of chromosome 2H$^{ch}$ on grain quality traits.

**Keywords:** grain colour; *Hordeum chilense*; wheat introgression; wheat quality; wild barley

## 1. Introduction

Narrow genetic diversity often limits the improvement of many traits in wheat. The introgression of genes from wild relatives to wheat has become a widely recognized genetic approach for increasing genetic diversity and, hence, the need to explore primary, secondary and tertiary gene pools of wheat has grown [1,2]. *Hordeum chilense* Roem. et Schultz. is a diploid wild barley that exhibits advantageous agronomic and quality characteristics [3–6]. Furthermore, its high crossability with other species of the tribe Triticeae, such as both durum and common wheat, [3–5] make it useful in cereal breeding.

Addition and substitution lines of alien chromosomes in a common wheat background, are useful for introgressing alien chromosomal segments carrying genes of agronomical interest into wheat. Chromosome addition lines of *H. chilense* in the *Triticum aestivum* L. cultivar "Chinese Spring" have been obtained including five for chromosomes 1H$^{ch}$, 4H$^{ch}$, 5H$^{ch}$, 6H$^{ch}$ and 7H$^{ch}$ and the ditelosomic addition line for the short arm of chromosome 2H$^{ch}$ [7]. Fertile wheat lines carrying deletions and translocations

involving chromosome 3H<sup>ch</sup> from *H. chilense* have also been obtained [8]. However, no addition or substitution lines in a common wheat background have been developed for chromosome 2H<sup>ch</sup>.

The location of agronomic traits on specific *H. chilense* chromosomes have been carried using these available wheat—*H. chilense* addition and substitution lines, such as resistance to greenbug (*Schizaphis graminum* Rond.) [9] and endosperm prolamins located on chromosome 1H<sup>ch</sup> [10,11]; resistance to *Septoria tritici* on chromosome 4H<sup>ch</sup> [12]; tolerance to salt on chromosomes 1H<sup>ch</sup>, 4H<sup>ch</sup> and 5H<sup>ch</sup> [13]; fertility restoration on chromosome 6H<sup>ch</sup> [14]; and carotenoid content on chromosome 7H<sup>ch</sup> [15]. Wheat—*H. chilense* translocation or recombinant lines have also been generated using both addition and substitution lines of *H. chilense* chromosomes in wheat background [8,11,16–18].

Chromosome 2H<sup>ch</sup> has the potential to improve seed carotenoid content in wheat. Genetic studies of yellow pigment content (YPC) in *H. chilense* revealed that chromosome 2H<sup>ch</sup> showed a significant association with YPC [19] and four endosperm carotenoid-related genes have been genetically mapped to chromosome 2H<sup>ch</sup>, such as geranyl geranyl pyrophosphate synthase (*Ggpps1*) for geranylgeranyl diphosphate synthesis, zeta-carotene desaturase (*Zds*), beta-carotene hydroxylase 3 (*Hyd3*) from the carotenoid biosynthetic pathway and polyphenol oxidase 1 gene (*Ppo1*) implicated in plant tissue enzymatic browning [20].

Molecular markers that are able to distinguish *H. chilense* chromosome 2H<sup>ch</sup> in wheat background provide a useful tool for selection. The conserved orthologous set (COS) [21] represents an important reservoir of markers that allow comparative studies with wheat and barley and their transference to *H. chilense* is a main goal.

The aims of this work were the following: (a) to obtain wheat—*H. chilense* chromosome 2H<sup>ch</sup> introgression lines; (b) to characterize the lines obtained by fluorescence *in situ* hybridization (FISH) and chromosome-specific simple sequence repeat (SSR) markers; (c) to transfer COS markers to *H. chilense* and to determine their arm location within 2H<sup>ch</sup> and (d) to compare the arm location with wheat and barley homoeologous group 2.

## 2. Materials and Methods

### 2.1. Plant Material

A "Chinese Spring" (CS) wheat—*H. chilense* double 2H<sup>ch</sup>(2D)-7H<sup>ch</sup>(7D) disomic substitution line, previously obtained at the University of Córdoba (results not shown), was used for inducing structural changes in chromosome 2H<sup>ch</sup> using gametocidal chromosome 2C<sup>c</sup> from *Aegilops cylindrica* host. The double 2H<sup>ch</sup>(2D) and 7H<sup>ch</sup>(7D) disomic substitution line was obtained by pollinating tritordeum (the fertile amphiploid between *H. chilense* and *T. turgidum* L., AABBH<sup>ch</sup>H<sup>ch</sup>, $2n = 6x = 42$) with a wheat disomic addition line for gametocidal chromosome 2C<sup>c</sup> from *Ae. cylindrica* Host. following the breeding procedure described in [8]. The double substitution 2H<sup>ch</sup>(2D) and 7H<sup>ch</sup>(7D) line was pollinated with the wheat disomic addition line for the gametocidal chromosome 2C<sup>c</sup> from *Ae. cylindrica*. The F1 plants monosomic for 2H<sup>ch</sup>, 7H<sup>ch</sup> and 2C<sup>c</sup> were selfed for four generations.

### 2.2. Fluorescence In Situ Hybridization (FISH)

The excised root tips were pretreated with ice water for 24 h and then fixed in acetic ethanol: acetic acid (3:1, v/v), as described previously [8]. The FISH protocol was carried out as described by [22]. The pAs1 sequence (1 kb) isolated from *Aegilops tauschii* Coss. [23] and *H. chilense* genomic DNA were used as probes. The pAs1 probe hybridizes to D-genome chromosomes of wheat [24] and H<sup>ch</sup>-genome chromosomes from *H. chilense* [25]. The pAs1 probe and *H. chilense* DNA were labeled with biotin-16-dUTP (Roche Diagnostics, Switzerland) and with digoxigenin-11-dUTP (Roche Diagnostics, Switzerland), respectively, by nick translation. Three plants per each introgression line were analyzed.

Biotin- and digoxigenin-labelled probes were detected with streptavidin-Cy3 conjugates (Sigma, St. Louis, MO, USA) and antidigoxigenin FITC (Roche Diagnostics) antibodies, respectively. The

chromosomes were counterstained with DAPI (4′,6-diamidino-2-phenylindole) and mounted in Vectashield mounting medium (Vector laboratories, Inc., Burlingame, CA, USA). A Leica DMR epifluorescence microscope was used for signal visualization. Images were captured with a Leica DFC7000T camera and processed with LEICA application suite v4.0 software (Leica, Germany).

### 2.3. Molecular Marker Analysis

A total of 82 COS markers from wheat homoeologous group 2 [21] were studied for their utility in *H. chilense* (File S1). *H. chilense* (line H7) and common wheat CS were used as controls. The CTAB method [26] was used for DNA extraction of young leaf tissue. The concentration of each sample was estimated using a NanoDrop 1000 Spectrophotometer (Thermo Scientific, Waltham, MA, USA). Amplifications were made using a TGradient thermocycler (Biometra, Göttingen, Germany) with 60 ng of template DNA in a 25 µl volume reaction containing 5 µl of 10× PCR Buffer, 0.5 µM of each primer, 1.5–2.0 mM $MgCl_2$, 0.3 mM dNTPs and 0.25 U of Taq DNA polymerase (BIOTOOLS B&M Laboratories, Madrid, Spain). The PCR conditions of COS markers were as follows: 4 min at 94°C, followed by 35 cycles of 45 s at 94°C, 50s at 58°C annealing temperature, 50 s at 72°C, and a final extension step of 7 min at 72 °C.

In addition, four chromosome-specific SSR markers for the wheat D-genome were used for molecular characterization of the introgression lines [27,28]. *Xgwm261* and *Xgwm157* markers were used to detect 2DS and 2DL chromosome arms, respectively. *Xcfd66* and *Xbarc111* were used to detect 7DS and 7DL chromosome arms, respectively. Amplifications were carried out as described at GrainGenes [29] One plant from each introgression line was used for the molecular characterization. "Chinese Spring", *H. chilense*, a ditelosomic 2H$^{ch}$S line, a ditelosomic 7H$^{ch}$S line, a ditelosomic 7H$^{ch}$L line and disomic substitution line CS 7H$^{ch}$(7D) were used as controls. Ditelosomic 2H$^{ch}$S and ditelosomic 7H$^{ch}$S lines were provided by the John Innes Centre (UK). The ditelosomic 7H$^{ch}$L line and disomic CS 7H$^{ch}$(7D) substitution line were obtained previously [17].

The amplified products were resolved using 2% agarose gels (SSRs) or polyacrylamide gels (10%, w/v; C: 2.67%) (COS) and stained with ethidium bromide or SafeView Nucleic Acid Stain (NBS Biologicals, Huntingdon, UK) incorporated in the gel. A 100 bp DNA ladder (Solis BioDyne, Tartu, Estonia) was used as a standard molecular weight marker. Kodak Digital Science 1D software (version 2.0) was used to determine the amplicon lengths.

### 2.4. Comparative Mapping

The orthologous relationship between the 2A, 2B, and 2D genome chromosomes of bread wheat and the 2H$^{ch}$ chromosome from *H. chilense* has been studied from the genomic perspective of wheat as described previously [30]. For the construction of the physical map, the expressed sequence tag (EST) source sequences (File S2) were used as queries in BLASTn searches against the wheat reference pseudomolecules [31] to identify the start positions (bp) of the ESTs. In this study, BLAST hits with $E$ values smaller than $1e^{-10}$, identity % > 58.44 and alignment length > 100 bp were considered significant. The genomic start positions in bp of the best hits in wheat pseudomolecules (File S3) were used to construct a physical map of the polymorphic COS markers. The wheat reference genome sequence [31] was used to determine the centromere positions for 2A, 2B and 2D wheat chromosomes. Both the length in bp of wheat pseudomolecules, as well as the start genomic positions of the ESTs, were converted to pixels. Then, the data from the BLASTn searches were used to construct a physical map for 2A, 2B, and 2D wheat chromosomes showing the position of the source EST of the COS markers assigned to *H. chilense* chromosome 2H$^{ch}$.

The rice locus (RAP) [21] was used to locate the COS markers in the barley genome zipper [32]. The RAP locus identifier was retrieved using the ID Converter tool [33]. The full-length barley cDNA corresponding to each rice locus was used for determination of the barley Unigene corresponding to each COS marker. The Unigene sequences were aligned in Barleymap [34] to obtain their positions in the International Barley Sequencing Consortium map [32,35].

## 3. Results

*3.1. Cytogenetic and Molecular Characterization of Wheat—H. chilense Introgression Lines Involving Chromosome 2H$^{ch}$*

The pAs1 and *H. chilense* genomic DNA used as probes in FISH analysis allowed the identification of a pair of 2H$^{ch}$ chromosomes and the absence of the wheat 2D chromosome pair in one line with 42 chromosomes. This result indicated that this line was disomic for the substitution 2H$^{ch}$(2D) (Figure 1a). The absence of 2D was tested using *Xgwm261-2DS* (Figure 2a) and *Xgwm157-2DL* (Figure 2b) molecular markers. A pair of telocentric chromosomes was identified by FISH in one line with 42 + 2t chromosomes (Figure 1b). To determine the chromosome arm involved in each introgression line, we used the *c749557* COS marker mapped on the 2H$^{ch}$S arm and *c731690* mapped on 2H$^{ch}$L, respectively (see Section 3.2). The presence of the c731690 marker for 2H$^{ch}$L and the absence of the *c749557* marker for 2H$^{ch}$S showed that this line was ditelosomic for the 2H$^{ch}$L arm (Figure 3a,b).

FISH analysis revealed a line apparently carrying chromosome 2H$^{ch}$ (Figure 1c). Marker *c731690* for 2H$^{ch}$L was amplified in this line, but there was no amplification of the *c749557* marker for 2H$^{ch}$S (Figure 3a,b). These results suggested that a 2H$^{ch}$L·2H$^{ch}$L isochromosome was present in this line, and it was named Iso 2H$^{ch}$L. Both the ditelosomic 2H$^{ch}$L and Iso 2H$^{ch}$L lines were nullisomic for chromosome 2D, as demonstrated by the absence of amplification of both *Xgwm261-2DS* (Figure 2a) and *Xgwm157-2DL* (Figure 2b) molecular markers.

We identified two lines carrying centromeric translocations involving the 2H$^{ch}$S chromosome arm and wheat chromosomes. One of these lines was homozygous for the 2H$^{ch}$S·2DL translocation (Figure 1d). The other translocation line was a double monosomic for 2H$^{ch}$S·2DL and 7H$^{ch}$S·D translocations (Figure 1e). Chromosome-specific SSR markers confirmed the absence of 2DS (Figure 2a) and the presence of 2DL (Figure 2b) in both lines. Amplification of the *c749557* marker (Figure 3a) and the absence of amplification of the *c731690* marker (Figure 3b) demonstrated the presence of 2H$^{ch}$S and the absence of 2H$^{ch}$L, respectively, in both lines. COS markers *c779791* and *c759439*, previously assigned to 7H$^{ch}$S and 7H$^{ch}$L, respectively [18], were used to detect introgression from chromosome 7H$^{ch}$ (Figure 3c,d). Amplification of the *c779791* marker specific for the 7H$^{ch}$S arm (Figure 3c) indicated the presence of 7H$^{ch}$S translocated to an unidentified wheat fragment. The presence of pAs1 signals on the wheat small fragment indicated that the chromosome 7H$^{ch}$S arm was translocated to an unidentified D-genome chromosome (Figure 1e). The absence of amplification of chromosome-specific markers *Xcfd66-7DS* and *Xbarc111-7DL* demonstrated the absence of a 7D chromosome pair in this line (Figure 2c,d).

In the remaining two lines, two centromeric translocations involving 2H$^{ch}$ and 7H$^{ch}$ *H. chilense* chromosomes were detected. One line was homozygous for the 7H$^{ch}$S·2H$^{ch}$L translocation (Figure 1f) and the other one was monosomic for the 2H$^{ch}$L·7H$^{ch}$L translocation. Both translocation lines were nullisomic for chromosome 2D (Figure 2a,b). Chromosome-specific SSR marker patterns for 2D (*Xgwm261-2DS* and *Xgwm157-2DL*) and 7D (*Xcfd66-7DS* and *Xbarc111-7DL*) genome chromosomes are given in Figure 2a,b and Figure 2c,d, respectively. Chromosome-specific marker results for chromosome 7H$^{ch}$S and 7H$^{ch}$L are given in Figure 3c,d. Table 1 shows the chromosome constitutions of all the *H. chilense* introgression lines. All lines were vigorous and seed set.

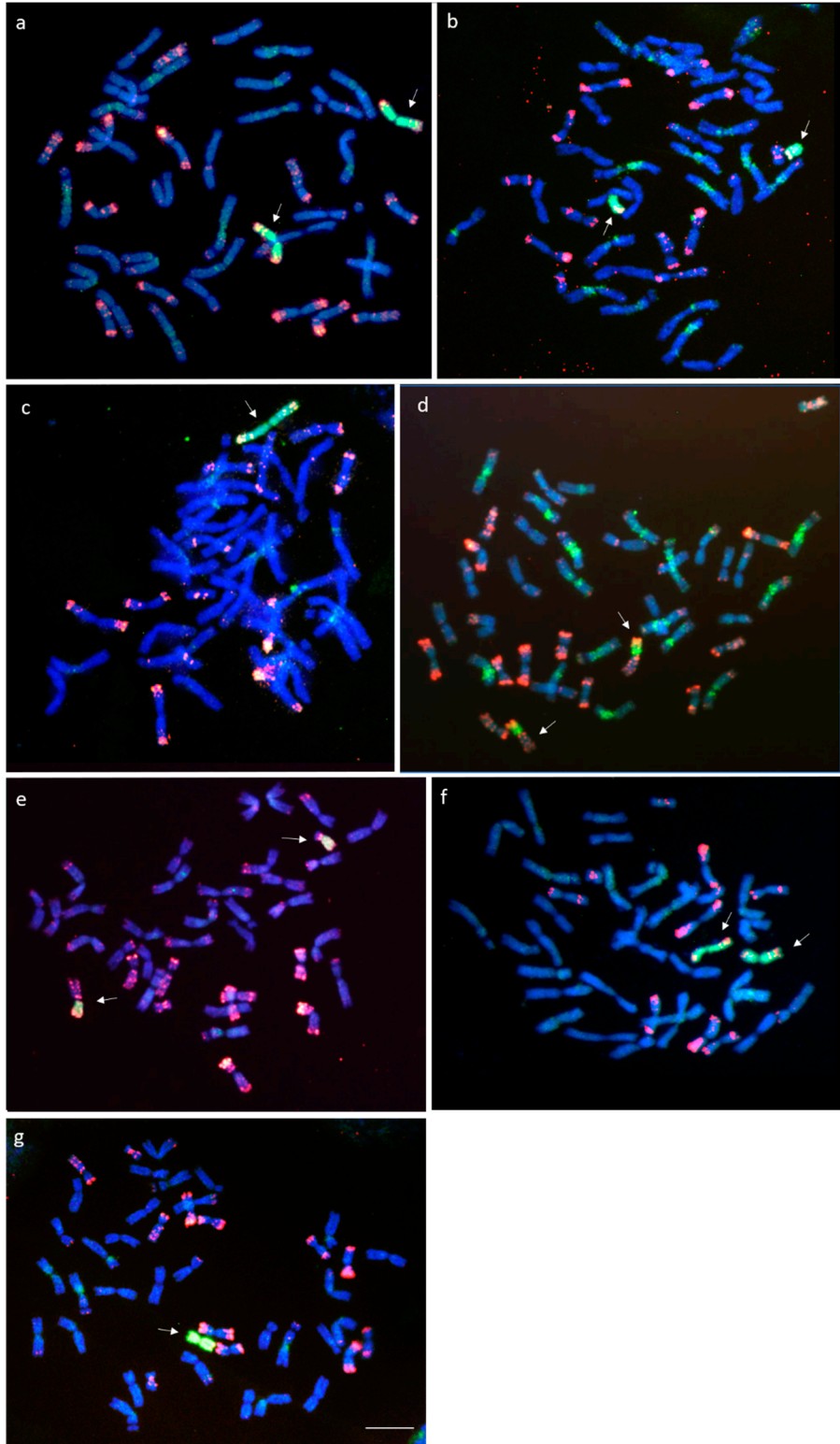

**Figure 1.** Fluorescence in situ hybridization (FISH) with the pAs1 repetitive (red) and *H. chilense* genomic DNA (green) probes to mitotic metaphase of wheat—*H. chilense* introgression lines involving chromosome 2H$^{ch}$. (**a**) Disomic substitution 2H$^{ch}$ (2D); (**b**) Ditelosomic 2H$^{ch}$L; (**c**) Isochromosome 2H$^{ch}$L; (**d**) Translocation 2H$^{ch}$S·2DL; (**e**) Translocation 2H$^{ch}$S·2DL + T7H$^{ch}$S·D; (**f**) Translocation 7H$^{ch}$S·2H$^{ch}$L; (**g**) Translocation 2H$^{ch}$L·7H$^{ch}$L. Bar = 10 μm.

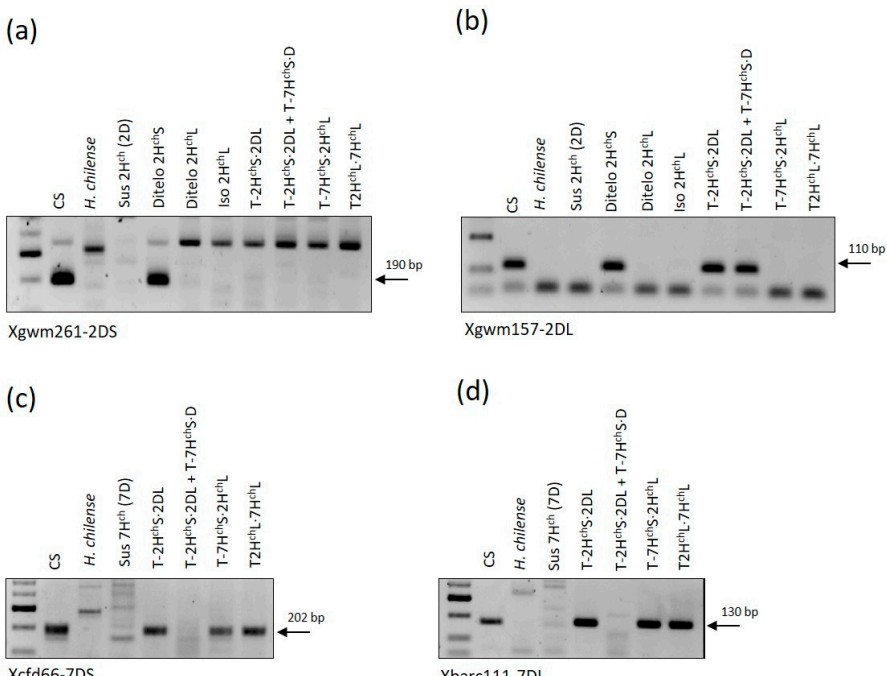

**Figure 2.** Molecular characterization of introgression lines with wheat chromosome-specific simple sequence repeats (SSR) markers. (**a**) *Xgwm261*-2DS; (**b**) *Xgwm157*-2DL; (**c**) *Xcfd66*-7DS and (**d**) *Xbarc111*-7DL.

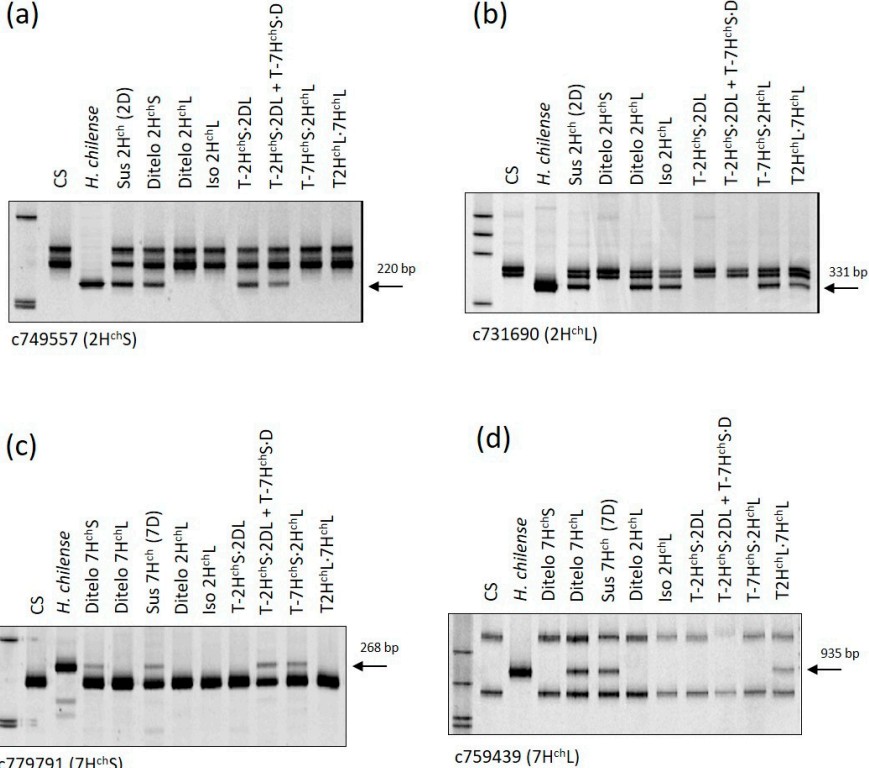

**Figure 3.** Examples of PCR amplification profiles used for identifying chromosome 2H$^{ch}$ and 7H$^{ch}$ arms in the introgression lines. (**a**) c749557 mapped on the short arm of chromosome 2H$^{ch}$; (**b**) c731690 mapped on the long arm of chromosome 2H$^{ch}$; (**c**) c779791 mapped on the short arm of chromosome 7H$^{ch}$; and (**d**) c759439 mapped on the long arm of chromosome 7H$^{ch}$. "Chinese Spring" (CS), *H. chilense* (H$^{ch}$), ditelo 2H$^{ch}$S, ditelo 7H$^{ch}$S, ditelo 7H$^{ch}$L and disomic substitution line CS 7H$^{ch}$(7D) were used as controls.

**Table 1.** Chromosome constitutions of wheat—*H. chilense* introgression lines involving chromosome 2H[ch].

| Line | Type of Aberration | *H. chilense* Introgressions | No. of D Chromosomes | No. of A/B Chromosomes | Total No. of Chromosomes |
|---|---|---|---|---|---|
| Sus 2H[ch] (2D) | Substitution [1] | 2H[ch] | 12 (2D pair absent) | 28 | 42 |
| Ditelo 2H[ch]L | Telosome [1] | 2H[ch]L | 12 (2D pair absent) | 30 | 42 + 2 telos |
| Iso 2H[ch]L | Isochromosome [2] | 2H[ch]L | 12 (2D pair absent) | 30 | 42 + iso |
| T2H[ch]S·2DL | Translocation [1] | 2H[ch]S | 12 + 2T (2DL absent) | 30 | 42 + 2T |
| T2H[ch]S·2DL + T7H[ch]S·D | Translocation [3] | 2H[ch]S+7H[ch]S | 8 + 2T (7D pair absent) | 28 | 36 + 2T |
| T7H[ch]S·2H[ch]L | Translocation [1] | 2H[ch]L+7H[ch]S | 9 (2D pair absent) | 30 | 39 + 2T |
| T2H[ch]L·7H[ch]L | Translocation [2] | 2H[ch]L+7H[ch]L | 12 (2D pair absent) | 30 | 42 + 1T |

[1] disomic; [2] monosomic; [3] double monosomic

### 3.2. Transferability and Chromosome Location of COS Markers in H. chilense

The transferability to *H. chilense* of 83 COS markers from wheat homoeologous group 2 was studied (File S1). First, all 83 markers were screened for polymorphisms (size polymorphisms or presence and absence) between *H. chilense* and common wheat. Of the 83 markers, 65 (78.3%) consistently amplified *H. chilense* products and 26 (40.0% of the total) were polymorphic between *H. chilense* and wheat (Table 2). Twenty-four of these 26 polymorphic markers were mapped to chromosome 2H[ch], as demonstrated by their presence in the wheat—*H. chilense* 2H[ch](2D) substitution line. We were unable to map the remaining two markers because they did not amplify products in any of the available wheat—*H. chilense* addition lines. Of the 24 COS markers mapped on chromosome 2H[ch], eight were located on 2H[ch]S and 16 were located on 2H[ch]L, as demonstrated by their presence and absence in 2H[ch]S or 2H[ch]L ditelosomic lines, respectively. Table 2 summarize the characterization and chromosome arm location of wheat COS markers on *H. chilense* chromosome 2H[ch]. Figure 3a,b shows examples of amplification of homoeologous group 2 COS markers.

**Table 2.** Characterization and chromosome localization of wheat conserved orthologous set (COS) markers on *H. chilense* chromosome 2H[ch].

| Marker | Product Size in *T. aestivum* | Product Size in *H. chilense* | Arm Location in *H. chilense* | Chromosome Location in Wheat [1] | Location in Wheat (cm) [1] | Location in Barley (cm) [2] |
|---|---|---|---|---|---|---|
| c723421 | 262–234 | 242 | 2H[ch]S | 2BS | 28.1 | 63.5 |
| c754613 | 775 | 750 | 2H[ch]S | 2AS-2BS-2DS | 31.7 | 46.3 |
| c745448 | 329 | 313–364 | 2H[ch]S | 2BS | 32.1 | 52.5 |
| c77095 | 997–1075 | 1186 | 2H[ch]S | 2AS-2BS-2DS | 35.9 | 47.7 |
| c741602 | 887 | 925 | 2H[ch]S | 2AS-2BS | 45.8 | 56.3 |
| c751379 | 889 | 911 | 2H[ch]S | 2AS-2BS | 56.3 | 59.2 |
| c733078 | 419 | 393 | 2H[ch]S | 2BS-2DS | 68.2 | Not found |
| c756234 | 432 | 311 | - | 2AS-2BS | 63.9 | 52.2 |
| c749557 | 251–268 | 220 | 2H[ch]S | 2AS-2DS | 69.0 | 59.2 |
| c740970 | 229 | 239 | 2H[ch]L | 2AL-2BL-2DL | 140.2 | 63.5 |
| c744070 | 204 | 221 | 2H[ch]L | 2AL-2BL-2DL | 143.7 | Not found |
| c729382 | 602 | 463 | 2H[ch]L | 2BL-2DL | 145.5 | 69.2 |
| c751189 | 805 | 851 | 2H[ch]L | 2AL-2BL-2DL | 145.8 | 70.54 |
| c749297 | 257 | 247 | 2H[ch]L | 2AL-2BL-2DL | 149.3 | 71.1 |
| BF291656 | 920 | 821 | 2H[ch]L | 2AL-2BL-2DL | 152.0 | 98.6 |
| c760814 | 554602 | | 2H[ch]L | 2DL | 167.4 | Not found |
| c747571 | 330 | 370 | 2H[ch]L | 2AL-2BL-2DL | 173.2 | 122.2 |
| c743473 | 252 | 276 | 2H[ch]L | 2AL-2BL-2DL | 173.5 | 82.75 |
| c779794 | 521–602 | 661 | 2H[ch]L | 2AL-2BL | 176.0 | 113.5 |
| c795564 | 573 | 629 | 2H[ch]L | 2AL | 178.5 | 113.5 |

**Table 2.** *Cont*.

| Marker | Product Size in *T. aestivum* | Product Size in *H. chilense* | Arm Location in *H. chilense* | Chromosome Location in Wheat [1] | Location in Wheat (cm) [1] | Location in Barley (cm) [2] |
|---|---|---|---|---|---|---|
| c730704 | 377 | 393 | 2H$^{ch}$L | 2DL | 179.4 | 156.7 |
| c741642 | 366-390 | 377 | 2H$^{ch}$L | 2BL-2DL | 187.3 | 126.0 |
| c746642 | 843–889 | 864 | 2H$^{ch}$L | 2AL-2BL-2DL | 189.5 | 156.7 |
| c747195 | 508 | 668 | 2H$^{ch}$L | 2AL-2BL-2DL | 190.2 | Not found |
| c731690 | 342–355 | 320 | 2H$^{ch}$L | 2BL-2DL | 191.7 | 102.8 |
| c756123 | 398–520 | 442 | - | 2AL-2BL-2DL | 198.2 | 136.8 |

[1] Quraishi et al. (2009); [2] Position in barley determined using Barleymap (http://floresta.eead.csic.es).

### 3.3. Wheat—H. chilense Group 2 Homoeology

To investigate wheat—*H. chilense* group 2 macrosyntenic relationships, the source ESTs of the 24 polymorphic COS markers mapped on chromosome 2H$^{ch}$ were BLASTed to the sequences of the wheat chromosomes [31]. All EST markers showed hits on wheat pseudomolecules (File S3). To produce a physical map (Figure 4), the start positions of the alignments of the best hits on the A, B, and D genomes were extracted. All markers assigned to 2H$^{ch}$S and 2H$^{ch}$L were located in the same arm of wheat homoeologous group 2 chromosomes (Table 2, Figure 4).

To determine the positions of COS markers in Barley-maps (Table 2) the align tool implemented in Barleymap [34] was used to determine the positions of COS markers in barley maps (Table 2). Four of the COS markers were not found in the barley genome zipper [32] so their position in barley could not be determined. A good correspondence between *H. chilense* and barley was found for the arm locations of COS markers (Table 2).

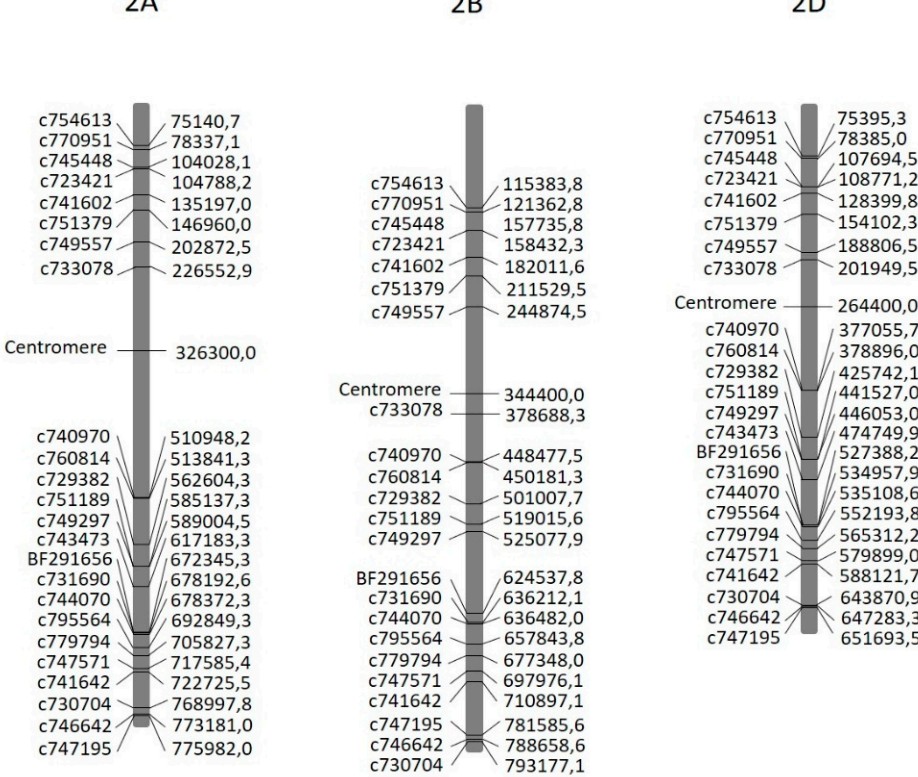

**Figure 4.** Visualization of wheat-*H. chilense* orthologous regions from the perspective of wheat homoeologous chromosome group 2. Physical map of the source expressed sequence tags (ESTs) of the COS-markers (left), the genomic positions on wheat pseudomolecules (kb) are on the right.

## 4. Discussion

A set of introgression lines involving chromosome 2H^ch from *H. chilense* in common wheat background was produced in this work using the gametocidal chromosome 2C^c from *Ae. cylindrica*. Gametocidal genes have been used to produce structural chromosome aberrations in wheat [36,37], barley [38,39] and rye [40]. In *H. chilense*, structural changes have been previously obtained for chromosomes 1H^ch, 3H^ch, 4H^ch and 7H^ch [8,11,17,18,41] and have been useful in determining the locations of genes and markers in this species. Breaks at both centromeric and interstitial regions of chromosomes have been induced by gametocidal chromosome 2C^c [36]. In the present study, telocentric and translocations between chromosome 2H^ch from *H. chilense* and wheat chromosomes have been generated

Alien addition and translocation lines are an ideal template for PCR-based mapping to assign molecular markers to chromosomes of the wild relatives of wheat [8,11,41–43]. Using gene-based conserved orthologous set (COS) markers on wheat—*H. chilense* introgression lines obtained in this work, we assigned a total of 24 markers to *H. chilense* chromosome 2H^ch. A 78.3% transference rate of COS markers to *H. chilense* chromosome 2H^ch was found. Since COS markers were intended for comparative studies among grasses, the high rate of transferability obtained in this work was expected [21]. Similar rates of transference of COS markers to *H. chilense* chromosome 7H^ch have been found previously [17]. COS markers have also been transferred successfully to other Triticeae species such as *Agropyron cristatum* [29,41] and *Aegilops* spp. [44].

The relevance of chromosome 2H^ch for endosperm carotenoid content has been highlighted by previous work. Association studies for YPC allowed the identification of three main chromosome regions for YPC variation in *H. chilense*, with the largest one located on chromosome 2H^ch and smaller regions detected on chromosomes 3H^ch and 7H^ch [20]. Four candidate genes associated with YPC were genetically mapped to chromosome 2H^ch: both *Ggpps1* and *Zds* were tightly linked and mapped near the centromere, while *Hyd3* and *Ppo1* were mapped to the long arm of chromosome 2H^ch [20]. Furthermore, a significant QTL at the distal part of chromosome 2H^ch has also been found where no carotenoid-related genes have been mapped [19,20]. The importance of chromosome 2H^chL in grain carotenoid content has also been revealed in the new cereal tritordeum (amphiploid derived from a cross between the wild barley *H. chilense* and durum wheat), which has higher carotenoid pigment content in its grain than durum or bread wheat [45,46].

Chromosome 7H^ch from *H. chilense* confers the capacity to accumulate higher carotene concentration in seeds [15]. The *Psy1* gene controlling the first step of the carotenoid biosynthetic pathway was mapped to 7H^chS [47]. Wheat—*H. chilense* chromosome 7H^ch introgression lines have been developed [17,18,48], and all the genetic stocks carrying *Psy1* from *H. chilense* show increased carotenoid content relative to common wheat [48,49]. The obtention in this work of translocation T7H^chS·2H^chL could be of interest for studying the effect of the *Psy1* gene located on the 7H^chS arm and both the *Hyd3* and *Ppo1* genes mapped to the long arm of chromosome 2H^ch [20]. Furthermore, the lines carrying T2H^chS·2DL and T2H^chS·2DL + T7H^chS·D translocations could be of interest for studying the effect of both the *Ggpps1* and *Zds* genes mapped to the short arm of 2H^ch, in the absence and presence of the *Psy1* gene from *H. chilense*, respectively.

The transference of desirable genes from wild relatives to wheat can be restrict by linkage drag and the lack of compensation for the wheat chromatin substituted. The identification of alien chromosomal regions carrying the genes of interest and the analysis of their homoeologous relationships with wheat chromosomes can overcome that difficulty. It has been pointed out that only translocations produced by homoeologous recombination are beneficial for wheat improvement [50,51]. In this work, wheat—*H. chilense* macrosyntenic comparisons using COS markers revealed that *H. chilense* chromosome 2H^ch exhibits good synteny with wheat homoeologous group 2 chromosomes. Comparative mapping of carotenoid-related genes mapped to chromosome 2H^ch also showed good collinearity between *H. chilense* and Triticeae species [20,46]. The 24 COS markers assigned to chromosome 2H^ch in this work will facilitate introgression of alien genes associated with this chromosome into wheat.

## 5. Conclusions

We used in situ hybridization and genotyping to characterize genetic stocks harboring chromosome 2H$^{ch}$ from *H. chilense* in a wheat background. As far as we know, no previous translocation lines involving chromosome 2H$^{ch}$introgressed into wheat have been described. The cytogenetic stocks developed here may constitute an important resource for studying the effect of chromosome 2H$^{ch}$ on wheat grain color. In addition, these cytogenetic stocks allowed the localization of a set of conserved orthologous set (COS) markers to specific arms in chromosome 2H$^{ch}$.The genomic position of orthologous unigene EST-contigs used for the COS marker design revealed a good macrosyntenic relationship between *H. chilense* chromosome 2H$^{ch}$ and wheat homoeologous group 2. The new wheat—*H. chilense* recombinations are useful for genetic studies and might also serve as a bridge for transferring genes associated with yellow pigment into a wheat background.

**Supplementary Materials:** The following are available online at http://www.mdpi.com/2073-4395/9/9/493/s1, File S1: COS markers used in this work together with their primer sequences, File S2: Source of the COS markers assigned to the *H. chilense* chromosome 2H$^{ch}$, File S3: Results of BLASTn search for COS markers assigned to *H. chilense* chromosome 2H$^{ch}$ in the reference sequences of hexaploid wheat group 2 chromosomes (www.wheatgenome.org/) and the start positions (bp) of the marker-specific ESTs.

**Author Contributions:** A.C. conceived and designed the study; C.P. performed the experiments; A.C. analysed the data and wrote the paper: all authors have read and approved the final manuscript.

**Funding:** This research was supported by grants AGL2014-53195-R and RTI2018-093367-B-I00 from the Spanish State Research Agency (Ministry of Science, Innovation and Universities), co-financed by the European Regional Development Fund (FEDER) from the European Union.

**Conflicts of Interest:** The authors declare no conflict of interest.

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
