# Peer review of "Development of wheat—Hordeum chilense Chromosome 2Hch Introgression Lines Potentially Useful for Improving Grain Quality Traits"

_agronomy, doi:10.3390/agronomy9090493_

Round 1
Reviewer 1 Report
The ms presents development of very unique wheat lines with introgressions from Hordeum chilense. These materials may have potential to improve wheat breeding towards increasing seed carotenoid content. The presented experiments were designed correctly and proper methods were implemented to generate results. In general ms is well written and results are organized clearly in tables and figures. Nevertheless, I noticed some minor problems:
1. Described experiments are not directly correlated with genes responsible for grain quality traits therefore I suggest to reword the title of the ms
2. In section Materials and Methods information about number of plants analysed per line should be added (taking into consideration both cytogenetic and molecular analysis).
3. Figure 1:
3.1. what points arrow 3 on picture 1b?
3.2. on picture 1d letter 'd' is missing
4. Table 1: on how many plants per line presented data were evaluated?
Author Response
We are grateful for corrections which clearly improve the manuscript. We have considered all the suggestions and included them in the revised version.
Bellow you can find the response to some specific comments.
1. Described experiments are not directly correlated with genes responsible for grain quality traits therefore I suggest to reword the title of the ms
We appreciate the comments of the reviewer. The final aim to obtain and characterize introgression lines involving chromosome 2Hch in wheat background is their use for improving seed carotenoid content in wheat, so we have changed the title of the work: Development of wheat-H.chilense chromosome 2Hch introgression lines potentially useful for improving grain quality traits.
2. In section Materials and Methods information about number of plants analysed per line should be added (taking into consideration both cytogenetic and molecular analysis).
We have included the number of plants evaluated in the M&M section, both in cytogenetic analysis as well as molecular analysis.
3. Figure 1:
3.1. what points arrow 3 on picture 1b?
3.2. on picture 1d letter 'd' is missing
We have corrected the two mistakes on Figure 1.
4. Table 1: on how many plants per line presented data were evaluated?
The number of plants evaluated on Table 1 is included in M&M section.
We hope the manuscript is now suitable to be published in Agronomy.

Reviewer 2 Report
The objectives of this study were the production of a set of wheat–H. chilense introgression lines using the gametocidal chromosome 2Cc from Ae. cylindrica, and their characterization by fluorescence in situ hybridization (FISH) and chromosome-specific simple sequence repeat (SSR) markers. Seven wheat – H. chilense introgression lines involving different types of aberrations (substitutions, telosomes, isochromosomes and translocations) of the chromosome 2Hch were obtained and adequately described and characterized. The wheat–H. chilense macrosyntenic comparisons using COS markers revealed that H. chilense chromosome 2Hch exhibits synteny with wheat homoeologous group 2 chromosomes. The examined cytogenetic stocks constitute an important resource for studying the effect of this chromosome on wheat, and for transferring useful agronomic traits into a wheat background.
The title adequately describe the subject of the manuscript and the abstract briefly tell what was done and summarize the main results and conclusions. The authors’ contribution is placed in its proper perspective in relation to the state of knowledge. The subject is developed logically and effectively and the manuscript is well organized and concise. The conclusions are adequate and supported by the data.
Author Response
We are very grateful for all the comments of the reviewer